# Assessing the Attitudes and Clinical Practices of Ohio Dentists Treating Patients with Dental Anxiety

**DOI:** 10.3390/dj4040033

**Published:** 2016-09-30

**Authors:** Kristin A. Williams, Sarah Lambaria, Sara Askounes

**Affiliations:** School of Dental Medicine, Department of Community Dentistry, Case Western Reserve University, 2124 Cornell Road, Cleveland, OH 44106, USA; sarahlambaria@gmail.com (S.L.); sea10@case.edu (S.A.)

**Keywords:** dental anxiety, dental fear, anxiolytics, nitrous oxide

## Abstract

Dental anxiety (DA) negatively affects patients’ oral and overall health. This study explored attitudes and clinical practices of licensed Ohio general dentists who treat patients with DA. **Methods:** An anonymous self-administered mail survey was sent to 500 general dentists licensed and practicing in Ohio. Responses to 21 pre-coded questions were analyzed. Frequencies were examined; cross-tabs, Chi-Square, and Fischer’s Exact Test were calculated for statements according to dentists’ gender. Alpha was set at *p* = 0.05. **Results:** Nearly all respondents treated anxious patients; males were more likely to find it challenging than females. Dentists were most familiar with distraction, although half found nitrous oxide to be an effective tool. Female dentists were more likely than males to be familiar with anxiolytics and find them effective. **Conclusion:** Overall, Ohio general dentists are most familiar with using distraction and nitrous oxide during appointments for anxious patients. Gender differences exist in attitudes towards anxiolytic use for patients with DA. Practice Implications: By identifying techniques that are comfortable for patient and practitioner, oral health disparities associated with DA may be reduced.

## 1. Background

Dental anxiety (DA) refers to a persistent, uncontrollable fear of the dentist, dental appointments and dental-related situations. “Dental anxiety” and “dental fear” are often used interchangeably [1]. DA has been studied since the 1960s [2], but despite advances in knowledge, the prevalence of DA has remained constant over the years [3]. An estimated 10% of individuals suffer from dental anxiety in a given population [4]. DA has been documented in several industrialized countries including Sweden and the UK [5,6]. A “vicious circle” of dental fear has been proposed to explain the mechanism by which DA is maintained [7]. By avoiding dental visits until problems can no longer be ignored, more invasive treatment is usually required, which may reinforce feelings of anxiety. Declining oral health may create discomfort and psychological distress, leading to feelings of shame or embarrassment, all of which may also accompany dental anxiety. In turn this may lead to future avoidance and further dental decline [7].

The maintenance of dental anxiety is problematic since DA has been linked to a number of negative oral health outcomes [6]. Patients with DA have worse oral health as compared to the non-anxious patients. Fearful patients have higher numbers of decayed tooth surfaces and missing teeth, and fewer filled teeth, compared to patients without DA [8]. DA is negatively associated with dental attendance; avoidance of the dentist puts patients with DA at additional risk for poor oral health outcomes [9,10,11]. Dental anxiety has negative repercussions for patients that extend beyond the walls of the dental office; not only does DA have detrimental health effects, but it also affects individuals’ cognitive and social well being [6].

Dentists can ease patients’ dental anxiety by choosing appropriate anxiety management techniques. A number of methods have been shown to reduce dental anxiety. Cognitive Behavioral Therapy, or CBT, involves psychotherapy addressing anxious cognitions [12,13]. CBT is a form of psychotherapy that treats problems by modifying dysfunctional emotions, behaviors and thoughts. It does not probe childhood wounds to get to the root cause of conflict, but instead CBT focuses on solutions. Behavior Therapy (BT) has also been shown to reduce anxiety for patients, and involves re-learning conditioned responses to stimuli such as the dental appointment [13]. Brief relaxation exercises and distraction using music are effective methods of anxiety reduction [14]. The tell-show-do method, in which the dentist tells the patient what is going to happen, shows the instruments and then performs the procedure is the most frequently used behavioral management practice for pediatric patients [15]. The use of nitrous oxide reduces anxiety in patients [16,17]. Premedication with anxiolytics is also supported for patients with DA [14]. The dentist must choose which anxiety reduction method to pursue, keeping in mind that some of these methods require specialized training or licensure.

Although a number of management techniques exist, little is known about utilization of these methods in general dental practice [18]. Previous studies suggest that male and female dentists may differ in their work practices and patterns [19,20]. This study sought to assess the attitudes and clinical practices of dentists in Ohio regarding dental anxiety, and to investigate possible gender differences.

## 2. Methods

This cross-sectional study consisted of an anonymous, self-completed mailed survey. The questionnaire was developed as a summer research project for dental students of Case Western Reserve University. It consisted of three sections. Section I contained demographic questions and asked about the setting in which participants practiced. Section II asked participants to indicate their agreement with 10 statements about DA, using a five-point Likert scale. Section III included four multiple-choice questions, asking about dentists’ preferred techniques for anxious patients. Participants were instructed to circle one or more responses. IRB exemption was received prior to beginning the project.

Five hundred dentists were invited to participate in the study. Participants were randomly selected from a list of licensed Ohio dentists. Only dentists currently practicing in the state were included. The mailing included a brief cover letter explaining the purpose of the survey. The cover letter included the definition: “Dental anxiety is a term used to describe an excessive fear of dental procedures, dental situations, and/or dental personnel”. It stressed the anonymity of the survey, and that responses would be aggregated. A pre-paid addressed return envelope was included. The surveys were mailed within an eight-week period.

Analysis was completed using Statistical Package for the Social Sciences (SPSS) Version 22. Frequencies were examined. For Section II, answers were recoded to a three-level Likert scale, and cross-tabs and Chi-Square tests were calculated for each item based on gender. For Section III, a new dichotomous variable was created for each response, where “1” indicated a circled response, and “0” indicated that it was not circled. Cross-tabs and Fischer’s Exact Test were computed to compare responses by gender. Alpha was set at *p* = 0.05.

## 3. Results

Out of the 500 surveys that were mailed, 145 surveys were returned (29.0% response rate). After excluding specialists from the sample, the sample size was *n* = 118 general dentists. The demographic characteristics of sampled dentists are shown in Table 1. The majority of the sample was male (79.1%). Most respondents were between 41 and 64 years old (63.6%), and 16.1% were over 65 years old. The majority graduated from Ohio dental schools: 59.3% attended The Ohio State University, and 23.7% attended Case Western Reserve University.

Ohio dentists’ practice patterns and attitudes about anxious patients are shown in Table 2. Dental anxiety was frequently recognized: 94% of respondents agreed that they treat patients with DA in their practices. The majority felt comfortable treating these patients, yet over 60% of respondents agreed that treating anxious patients was challenging. Over 70% of respondents scheduled extra time for appointments, and nearly 62% had referred a patient to a different provider due to his or her dental anxiety. Cost was not found to be a major deterrent of treating patients with dental anxiety, but stress prevented one-fourth of respondents from treating more patients with dental anxiety. Male dentists were significantly more likely to agree that treating anxious patients was challenging compared to female dentists (71.3% compared to 54.8%). This was the only statement in Section II for which significant gender differences were observed (*p* = 0.02).

Ohio dentists’ clinical preferences for specific techniques to manage dental anxiety are shown in Table 3. The majority of respondents was familiar with distraction (82.8%), and was comfortable using this technique (80.2%). Distraction was most frequently identified as effective compared to other techniques, with over half of respondents agreeing. The second most recognized technique was nitrous oxide, with 80.2% of respondents indicating that they were familiar with nitrous oxide. Fewer respondents were comfortable using this technique, and approximately half of respondents indicated it was effective. Three-fourths of respondents were familiar with tell-show-do; nearly as many were comfortable using it, and half of respondents found this technique effective. Just over half the sample was familiar with anxiolytics. Behavior Therapy (BT) was not commonly recognized, and few dentists (6.0%) found it to be an effective means of anxiety reduction. Cognitive Behavioral Therapy (CBT) was the least commonly recognized technique; only 3.4% of respondents found this technique effective. Some dentists (8.6%) preferred an unlisted technique; only one dentist indicated that none of the techniques were effective for managing DA.

Significant gender differences in managing dental anxiety were observed regarding anxiolytics. Respondents’ attitudes about anxiolytics by gender are shown in Table 4. Female dentists were significantly more likely to indicate that they were familiar with anxiolytics than male dentists (71.0% compared to 44.4%). Females were also more likely to report anxiolytics effective for managing DA compared to male dentists. Although female dentists were more likely to indicate that they were comfortable using anxiolytics, the difference did not reach statistical significance. These were the only statements in Section III for which gender differences were observed.

## 4. Discussion

Dental anxiety not only affects patients, it also affects dentists entrusted with their care. In this study a majority of dentists recognized DA in their own practice and felt comfortable treating these patients. The complexity of treating patients with dental anxiety was reflected in the survey responses as well; many respondents agreed that treating patients with DA was challenging, and some felt it was stressful. The increased frequency with which male providers agreed that treating anxious patients was challenging compared to female providers may reflect differences in the way that males and females perceive and handle stressful situations in the office. Further research is needed to address this.

The techniques that sampled dentists most frequently indicated feeling comfortable with included distraction, tell-show-do, and nitrous oxide. This does not mean that these are the techniques that are utilized most often in private practice, but it does convey the knowledge and preferences of the dentists in the sample. By extension, this study provides information regarding what techniques are likely employed in clinical practice. If a dentist does not feel comfortable with a technique, he or she will most likely not use it.

A majority of sampled dentists were familiar with distraction and felt comfortable using it; this is not surprising, since distracting a patient with idle conversation or music does not involve specialized training. Distraction is non-invasive to the patient and the practitioner incurs no additional expense. The distraction technique is taught in dental schools as a way to put patients of all ages at ease.

Nitrous oxide was the second most frequently recognized technique. This may be due to the relatively long history of nitrous oxide use for anxiety reduction compared to other techniques: nitrous oxide has been praised for its ability to relax dental patients since the 1970s [21]. Although nitrous oxide is non-invasive and fairly inexpensive, it is a form of sedation and therefore requires proper training, licensure and equipment [22]. These extra measures might account for the relatively lower proportion of dentists who were comfortable using nitrous oxide compared to distraction. According to the Dental Practice Act for the State of Ohio, dentists cannot administer conscious sedation without a permit. In order to receive a permit, Ohio dentists must complete a minimum of 60 h of didactic instruction and 20 cases of clinical experience commensurate with each intended route(s) of administration (oral for children 12 years or younger, non-intravenous, parenteral, or intravenous) [23].

Respondents were least familiar with Cognitive Behavioral Therapy and Behavioral Therapy for dental anxiety reduction. These methods involve psychological techniques, and are not a traditional part of dental school curricula. They are also the most time-consuming techniques included in the survey, requiring multiple sessions prior to the dental appointment [13]. Using CBT or BT requires dentists to seek additional training or assistance from a professional familiar with that method. These factors may account for the number of dentists who were unfamiliar with CBT and BT.

It is important to recognize that the dentists’ indication of “effective” techniques reflect their own perceptions of effectiveness. The intent of this survey was to gauge what dentists felt were the most effective techniques. Dentists more frequently identified some techniques as being “more effective” than others. These responses did not necessarily account for patients’ perceptions or actual reductions in short- or long-term DA. For example, only half of the sampled dentists perceived that nitrous oxide was effective. One might expect more dentists to find this technique effective, given that the level of anxiety experienced by a patient with DA can be reduced to that of a non-anxious patient using nitrous oxide [22]. Future surveys might examine what factors general dentists take into account when determining whether they feel that a specific technique is effective.

This study found that female dentists were more likely than male dentists to be familiar with anxiolytics and to find them effective for patients with DA. Gender differences in the prescription of anxiolytics may reflect the increased likelihood of women to obtain psychotropic drugs compared to men, which is well established [24,25]. Gender differences in female dentists’ use of anxiolytics for patients may largely reflect cultural patterns of use. Future research may shed light on this difference.

The authors recognize the limitations of this study. Cross-sectional studies represent the views of one sample at a specific point in time. This suggests the sample may not be generalizable beyond Ohio; dentists from different regions may have different practice patterns and preferences for managing DA. Additionally, the self-administered survey may suffer from reporting bias: some dentists may have attempted to provide socially desirable responses. Further limitations of this study include a low response rate and small sample size. Despite these limitations, this study provides information about Ohio dentists’ attitudes and clinical practices regarding dental anxiety, which are not well known. These findings may guide future research.

Additional research is needed about dentists’ clinical practices for anxious patients. Continued training for dentists may build skills and increase their confidence and competence when treating patients with dental anxiety. Patients with DA have poor oral health outcomes and decreased dental care compared to their peers; identifying treatment methods that are comfortable for both patient and practitioner may decrease oral health disparities associated with DA.

## 5. Conclusions

Dental anxiety is commonly encountered in a general dental practice. Nearly all sampled Ohio general dentists stated that they have treated patients with dental anxiety, and the majority agreed that doing so is challenging. Male dentists were more likely to perceive treating anxious patients as challenging compared to female dentists. General dentists found using distraction, nitrous oxide, and tell-show-do to be the most effective techniques for treating patients with DA, although opinions about methods of anxiety reduction remain somewhat mixed. Gender differences exist in attitudes about anxiolytic use for anxious patients. Future studies should review effective dental anxiety reduction techniques from the patient’s perspective.

## 6. Practice Implications

A variety of techniques are available to help dental providers treat patients with dental anxiety. Many dentists schedule extra time for anxious patients, which may facilitate a successful appointment. Dentists are encouraged to find a management technique that is comfortable and effective for them. Dental providers interested in learning more about sedation may reference the American Dental Association’s Guidelines for the Use of Sedation and Anesthesia [26]. Dental schools need to address the issue of treating patients with dental anxiety further while training future practitioners. Accessible continuing education classes on the use of anxiolytics are needed to encourage more practitioners to treat the patient with dental anxiety.

## Figures and Tables

**Table 1 dentistry-04-00033-t001:** Demographics of the study sample.

Total	*n* = 118	
	*n*	%
**Age (*n* = 118)**		
20–40 years	24	20.3
41–64 years	75	63.6
65+ years	19	16.1
**Gender (*n* = 114)**		
Male	82	71.9
Female	32	28.1
**Dental School Alma Mater (*n* = 116)**		
The Ohio State University	70	60.3
Case Western Reserve University	28	24.1
Other (Out of State)	18	15.6

**Table 2 dentistry-04-00033-t002:** Ohio dentists’ practice patterns and attitudes about patients with dental anxiety.

Statement	Strongly Disagree No. (%)	Disagree No. (%)	Neither Agree Nor Disagree No. (%)	Agree No. (%)	Strongly Agree No. (%)
I treat patients with dental anxiety (DA) in my practice (*n* = 117)	1 (0.9)	1 (0.9)	5 (4.3)	57 (48.7)	53 (45.3)
I feel comfortable treating patients who experience DA (*n* = 117)	1 (0.9)	9 (7.7)	17 (14.5)	57 (48.7)	33 (28.2)
I find it challenging to treat patients with DA (*n* = 115)	4 (3.5)	12 (10.4)	21 (18.3)	57 (49.6)	21 (18.3)
I find it rewarding to treat patients with DA (*n* = 117)	6 (5.1)	9 (7.7)	31 (26.5)	50 (42.7)	21 (17.9)
I modify the treatment plan if I know that a patient has DA (*n* = 116)	9 (7.8)	21 (18.1)	26 (22.4)	49 (42.2)	11 (9.5)
I schedule extra time for appointments if I know a patient has DA (*n* = 115)	2 (1.7)	11 (9.6)	12 (10.4)	63 (54.8)	27 (23.5)
I have referred a patient to a different dental provider before because of his/her dental anxiety (*n* = 117)	15 (12.8)	23 (19.7)	7 (6.0)	54 (46.2)	18 (15.4)
I would treat more patients with DA, but it is too time consuming (*n* = 117)	18 (15.4)	40 (34.2)	37 (31.6)	16 (13.7)	6 (5.1)
I would treat more patients with DA, but it is too costly (*n* = 117)	18 (15.4)	41 (35.0)	39 (33.3)	16 (13.7)	3 (2.6)
I would treat more patients with DA, but it is too stressful (*n* = 115)	14 (12.2)	38 (33.0)	33 (28.7)	24 (20.9)	6 (5.2)

**Table 3 dentistry-04-00033-t003:** Ohio dentists’ clinical preferences for techniques used to treat patients with dental anxiety.

Statement	Familiar with Technique No. (%)	Comfortable with Technique No. (%)	Find Technique Effective No. (%)
Relaxation	49 (42.2)	49 (42.2)	34 (29.3)
Behavior Therapy (BT)	20 (17.2)	17 (14.7)	7 (6.0)
Cognitive Behavioral Therapy (CBT)	14 (12.1)	6 (5.2)	4 (3.4)
Distraction (music, talking, etc.)	96 (82.8)	93 (80.2)	64 (55.2)
Tell-show-do	87 (75.0)	84 (72.4)	57 (49.1)
Anxiolytics	61 (52.6)	51 (44.0)	47 (40.5)
Nitrous oxide or sedatives	93 (80.2)	76 (65.5)	58 (50.0)
Other	9 (7.8)	7 (6.0)	10 (8.6)
None of the above	2 (1.7)	2 (1.7)	1 (0.9)

**Table 4 dentistry-04-00033-t004:** Gender differences in Ohio dentists’ familiarity, comfort, and belief in efficacy of anxiolytics for the relief of dental anxiety.

Statement	Males (%)	Females (%)	*p*-Value
Familiar with Anxiolytics	44.4	71.0	0.02
Comfortable Prescribing Anxiolytics	37.0	58.1	0.06
Find Anxiolytics Effective	33.3	54.8	0.05

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
