# Peer review of "Assessing the Attitudes and Clinical Practices of Ohio Dentists Treating Patients with Dental Anxiety"

_dentistry, 2016, doi:10.3390/dj4040033_

Round 1

Reviewer 1 Report

SEE ATTACHED FILE 

Author Response

L37-44 Paragraph needs re-writing, refs missing.

             Background: Second paragraph was reworded with references added.

L46-47 & L47-48 define CBT better and define BT

            Bot CBT and BT were clearly defined. References were added.

L52 ref is irrelevent- Tell Show do needs justification

           Tell-show-do is defined and justified with new pediatric journal reference added. Previous 

           reference is removed.

L53 Old N2O reference is irrelevant

            Reference has been changed and N2O statement rephrased

L58-59 & L106-115 Since this is only Ohio... & Ohio's required training?...          

            Added statement from Ohio Dental Practice Act from the State Dental Board addressing p

            permit requirements and education/training requirements for N2O use. Reference cited.

L204-214 References are missing

            Reference list corrected and amended to reflect above changes.

Attached is the revised manuscript with changes tracked. 

Reviewer 2 Report

Dear Auhtors;

I think the manuscript is well written. However, minor revisions are required including:

In introduction:

The maintenance of dental anxiety is problematic since DA has been linked to a number of negative outcomes:  explain more about what negative outcomes??

In discussion:

Dental anxiety not only affects patients, it affects dentists entrusted with their care :

it should change to:   it also affects dentists ........

last paragraph of discussion:

and decreased dental utilization compared to their peers:  dental utilization should change to dental care

this sentence should be corrected : Male dentists were more likely to perceive treating anxious patients as a challenge as compared to female dentists :

 correction: needs to be change to aschallenging as......

Thank you

Author Response

Revisions were completed in the introduction and discussion as noted.

Attached is the manuscript with changes tracked.Thank-You.
